# Structure of the native pyruvate dehydrogenase complex reveals the mechanism of substrate insertion

Jana Škerlová [1], Jens Berndtsson [1], Hendrik Nolte[2], Martin Ott [1,3✉] & Pål Stenmark [1,4✉]

The pyruvate dehydrogenase complex (PDHc) links glycolysis to the citric acid cycle by converting pyruvate into acetyl-coenzyme A. PDHc encompasses three enzymatically active subunits, namely pyruvate dehydrogenase, dihydrolipoyl transacetylase, and dihydrolipoyl dehydrogenase. Dihydrolipoyl transacetylase is a multidomain protein comprising a varying number of lipoyl domains, a peripheral subunit-binding domain, and a catalytic domain. It forms the structural core of the complex, provides binding sites for the other enzymes, and shuffles reaction intermediates between the active sites through covalently bound lipoyl domains. The molecular mechanism by which this shuttling occurs has remained elusive. Here, we report a cryo-EM reconstruction of the native *E. coli* dihydrolipoyl transacetylase core in a resting state. This structure provides molecular details of the assembly of the core and reveals how the lipoyl domains interact with the core at the active site.

[1] Department of Biochemistry and Biophysics, Stockholm University, Stockholm, Sweden. [2] Max-Planck-Institute for Biology of Ageing, Cologne, Germany. [3] Department of Medical Biochemistry and Cell Biology, University of Gothenburg, Gothenburg, Sweden. [4] Department of Experimental Medical Science, Lund University, Lund, Sweden. ✉email: martin.ott@dbb.su.se; stenmark@dbb.su.se

The pyruvate dehydrogenase complex (PDHc) is a multi-enzyme complex of megadalton size that converts pyruvate into acetyl-coenzyme A (Fig. 1a, b), thereby linking glycolysis to the citric acid cycle and to the biosynthesis of fatty acids and steroids[1]. It is a central metabolic gatekeeper crucial for glucose homeostasis under different metabolic conditions in mammals[2]. Consequently, it plays roles in several diseases including obesity, type 2 diabetes, neurodegenerative disorders, and several types of cancer[3,4]. PDHc consists of multiple copies of three enzymes: pyruvate dehydrogenase (E1p; EC 1.2.4.1), dihydrolipoyl transacetylase (E2p; EC 2.3.1.12), and dihydrolipoyl dehydrogenase (E3; EC 1.8.1.4)[5]. It is structurally and functionally related to the other oxoacid dehydrogenase complexes: α-ketoglutarate dehydrogenase complex (E1o, E2o, and E3) and branched-chain α-ketoacid dehydrogenase complex (E1b, E2b, and E3), which all have individual E1 and E2 subunits but share a common E3 component[6]. Mitochondrial PDHc contains an E3-binding protein (protein X), and its activity is additionally regulated by E1 kinases and phosphatases[7–9]. In the first reaction step catalyzed by E1p, pyruvate is decarboxylated using thiamine pyrophosphate (TPP, also often referred to as TDP or ThDP for thiamine/vitamin B1 diphosphate) as a cofactor and the resulting hydroxyethyl-TPP then reacts with lipoamide (oxidized lipoic acid attached to a lysine residue of E2p[10]) to produce

S-acetyldihydrolipoamide and recover the TPP. In the second step, E2p catalyzes the reaction of S-acetyldihydrolipoamide with coenzyme A (CoA), yielding dihydrolipoamide and the final product acetyl-CoA. Dihydrolipoamide is recycled into the oxidized form in the third step by flavin adenine dinucleotide (FAD)-dependent E3 with oxidized nicotinamide adenine dinucleotide ($NAD^+$) acting as the final electron acceptor (see Fig. 1a).

The size, composition, and symmetry of the oxoacid dehydrogenase complexes vary in different organisms, but the structural and functional core is always a dihydrolipoyl acyltransferase (E2). It provides binding sites for the other enzymes (together with the E3-binding proteins in mitochondria[9,11]) and conveys reaction intermediates between spatially distant active sites in the complex. The covalent substrate channeling conducted by E2 (and E3-binding protein in eukaryotes) tightly couples the individual reactions and increases the overall rate of the whole complex[1,3,6,7] (Fig. 1f). E2 is a multidomain protein that comprises up to five domains connected through flexible linkers: one to three tandem lipoyl domains followed by one peripheral subunit-binding domain, and one catalytic domain on the C-terminus[12–16] (Fig. 1e). In Gram-negative bacteria the E2p catalytic domains are arranged in eight tight trimers further associated into a 24-meric cubic structure with an E2p trimer in each corner of the cube with a 432 symmetry[6,17,18]. The

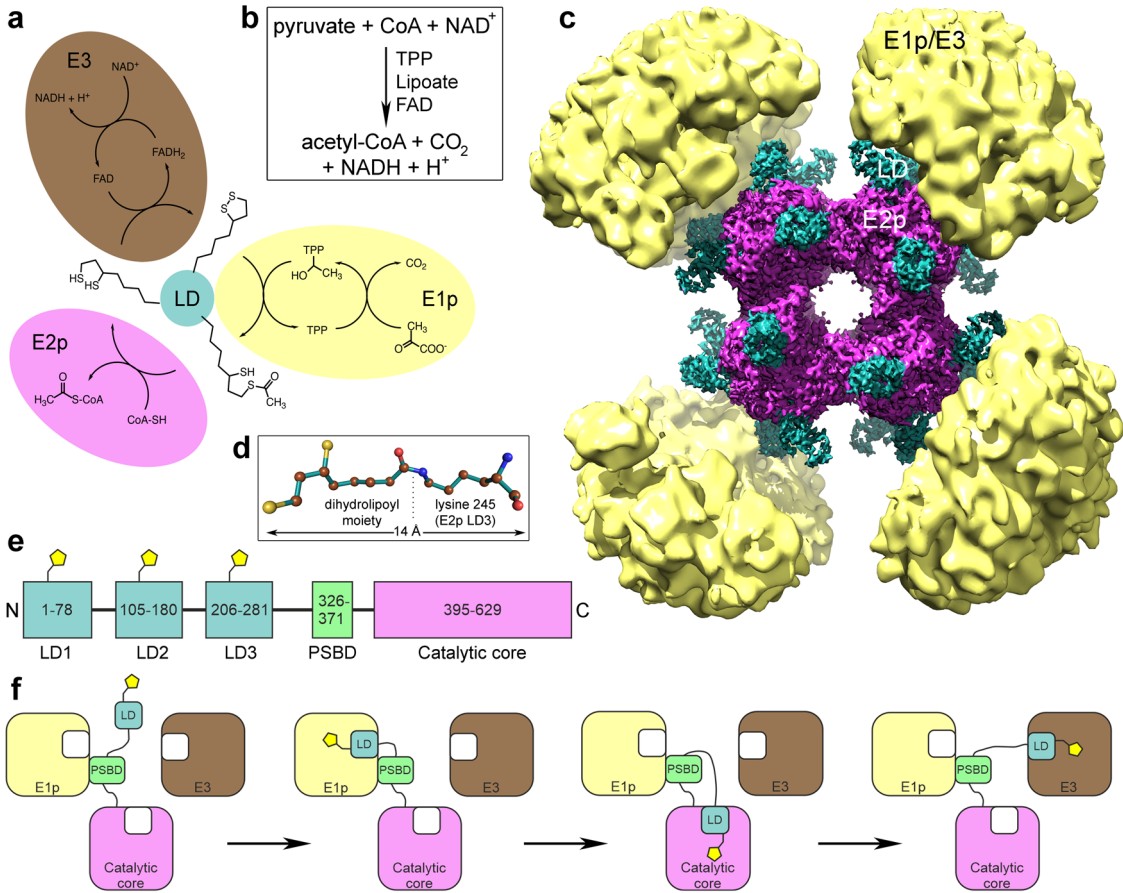

**Fig. 1 Reaction mechanism and cryo-EM reconstruction of *E. coli* PDHc. a** Scheme of the reactions catalyzed by the individual components of PDHc. **b** Overall reaction catalyzed by PDHc. **c** High-resolution cryo-EM reconstruction of the inner E2p core in alignment with the low-resolution reconstruction of the outer shell of the complex. Catalytic cores of E2p are colored magenta, lipoyl domains cyan, and the outer shell components (E1p/E3) yellow. **d** Ball-and-stick representation of the dihydrolipoyllysine moiety. **e** Scheme of the domain organization of E2p. **f** A simplified scheme of the covalent swinging-arm substrate channeling mechanism in the catalytic cycle of PDHc. Only one LD is shown and only one copy of each enzyme is shown, but the LDs can reach multiple copies of each enzyme in the multienzyme complex. PSBD can bind to either E1p or E3. E1p—pyruvate dehydrogenase, E2p—dihydrolipoyl transacetylase, E3—dihydrolipoyl dehydrogenase, LD—lipoyl domain, PSBD—peripheral subunit-binding domain. Panels **d**–**f** were prepared based on Arjunan et al.[19].

peripheral subunit-binding domain recruits E1p and E3 into the PDHc[19], and these enzymes are located in the outer shell surrounding the core of the complex. Each lipoyl domain can carry one lipoic acid covalently attached to a lysine residue[20] (Fig. 1d), and they function as swinging arms channeling the substrate between the individual enzymes' active sites in the PDHc[8,21,22] (Fig. 1f). Each *E. coli* E2p has three N-terminal lipoyl domains, and therefore the core E2p 24-mer carries 72 lipoyl domains that participate in delivering covalently attached substrates to the different active sites in the complex.

Despite decades of structural studies on the PDHc[18,19,23–40], the molecular mechanism of the substrate channeling by lipoyl domains has remained elusive. We report here the cryo-EM reconstruction of the cubic E2p core of *E. coli* PDHc in a native resting state (in the absence of substrates), in which a lipoyl domain is bound to the catalytic domain of each E2p and the dihydrolipoyllysine residue is immerged deep in the E2p active site channel (coenzyme A is not present). This structure therefore reveals the interaction between the catalytic domain and the lipoyl domain of an E2 enzyme, and shows how the lipoyl domain likely inserts the substrate-carrying prosthetic group into the active site of the dihydrolipoyl acyltransferase.

## Results and discussion

We performed single-particle cryo-EM structural analysis on the native PDHc isolated from *E. coli* K12 cells (Supplementary Fig. 1a). The identity and intactness of the isolated complex was verified by mass spectrometry (Supplementary Table 1) and specific enzymatic tests (Supplementary Fig. 1b). The presence of the lipoamide modification was confirmed by a western blot analysis using an anti-lipoate antibody (Supplementary Fig. 2b) and by mass-spectrometry analysis (lipoylated residues K41, K144, and K245; Supplementary Fig. 3 and Supplementary Table 1). The *E. coli* K12 cells had a FLAG tag in the genome on the gene coding for the UbiF protein, as they were originally designed for isolation of the ubiquinone synthesizing complex, which was also present in low amounts in the purified PDHc sample (Supplementary Table 1). The isolation of the PDHc using the anti-FLAG resin was reproducible and independent of the presence of the FLAG epitope, as similar purification outcome was achieved with wild type *E. coli* K12 cells (Supplementary Fig. 2).

The cells were cultivated in minimal media supplemented with succinate and in the absence of glucose. Therefore only low amounts of pyruvate were available in the cells and the pyruvate dehydrogenase complex could only perform its catalytic function at a very limited capacity, and was therefore isolated in what we refer to as a resting state.

We obtained a low-resolution cryo-EM reconstruction of the whole native PDHc containing all subunits (approx. 4.5 MDa), including low-resolution map for the outer shell (Fig. 1c and Supplementary Fig. 4). We have also confirmed the presence of the outer shell components on the level of 2D classes, which were created using particles in which the signal for the E2p core had been subtracted (Supplementary Fig. 4d). Some of the 2D classes clearly exhibit the presence of the peripheral subunit-binding domains and/or lipoyl domains bound to the E1p and/or E3 components, as well as the linker segment (Supplementary Fig. 4e), which was also apparent in the reconstructed 3D volume, connecting the E2p catalytic core with the outer shell (Supplementary Fig. 4c). These E1p and/or E3 components associate into clusters in the area near the edges of the cubic E2p core (Supplementary Fig. 4b), in line with a previous cryo-electron tomography study[40]. However, we were not able to unambiguously assign the individual E1p and/or E3 components into a complete

model due to the conformational flexibility and compositional heterogeneity of the outer shell (Supplementary Fig. 4a), which is not geometrically precise and does not exactly follow the symmetry of the E2p core[6,8,9,28,29,40].

Therefore, we focused on the cubic E2p core, where we could impose an octahedral symmetry to achieve high resolution. Interestingly, already during initial 2D classifications, 24 "knobs" were clearly visible on the catalytic core. These protrusions were later identified as the lipoyl domains bound to the E2p catalytic domains (Fig. 1c and Supplementary Fig. 5). Since about 30% of the E2p particles lacked ordered lipoyl domains, we excluded them from the final 3D refinement (Supplementary Fig. 5c). We determined the cryo-EM structure of the cubic E2p core to a nominal resolution of 3.16 Å (FSC criterion of 0.143, Fig. 1c, Supplementary Fig. 5, and Supplementary Table 2). The local resolution of the E2p catalytic core domain ranged from 2.8 to 3.5 Å (Supplementary Fig. 5), while the resolution for the lipoyl domains was lower, ranging between 3.5 and 7.5 Å (values based on the FSC criterion of 0.5).

The overall structure of the E2p shows the expected cubic core (Fig. 2), consisting of trimers of the catalytic domains with the associated lipoyl domains, which bind to the catalytic domains in a manner that follows the overall octahedral symmetry of the entire complex. We used the X-ray structure of the catalytic domain (PDB 4n72[37]) and the NMR solution structure of the innermost lipoyl domain (PDB 1qjo[39]) as initial docking models. The three lipoyl domains of *E. coli* E2p are 73% identical and 95% similar in sequence, and differ significantly in only 4 out of 74 residues at positions further away from the site of interaction with the catalytic domain (Supplementary Fig. 6a). Based on the quality of the cryo-EM map, we cannot distinguish between the three individual lipoyl domains. Considering the octahedral symmetry and the functional mechanism of the multienzyme complex, where the lipoyl domains can act redundantly, randomly, and independently[1,41–44], even though the individual E2 chains might divide their catalytic duties so that their lipoyl domains only visit certain active sites in the neighborhood[45,46], the structure likely represents an averaged mixture of all three lipoyl domains.

Residue K245 carries a covalently attached dihydrolipoic acid through an amide bond (dihydrolipoamide, Fig. 1d). Given the catalytic mechanism of the complex and the very low availability of pyruvate during the growth of the cells, the prosthetic group could theoretically exist in both the reduced and oxidized forms in the resting state. In order to analyze the state of the lipoyllysine in the sample, we performed mass-spectrometry experiments and analyses, which indeed identified only a minor amount of the acetylated form of the lipoyllysine compared to the majority of lipoyllysine in the non-acetylated form (Supplementary Fig. 3). We were however not able to unambiguously identify the oxidation state of the lipoyllysine, and we expect that the resting state of the PDHc comprises a mixture of oxidized and reduced lipoyllysines.

The hydrophobic nature of the (dihydro)lipoyllysine moiety likely favors the bound state over a free unbound lipoyl domain. This might be especially pronounced because of the very high local concentration of the lipoyllysine resulting from the covalent tethering of the lipoyl domain. The reduced form may bind to both E2p and E3, while the oxidized form may bind to both E1p and E3 (Fig. 1a), and the reduced form likely has a higher affinity towards E2p than the oxidized form. Due to the lack of pyruvate in the cells, the PDHc exists in a resting state, where E1p is not active. Lipoylated LDs have been shown to bind to E1p (and also to E1o) with a lower affinity also in the absence of pyruvate[47] (resting state). This work[47] was done using only isolated domains

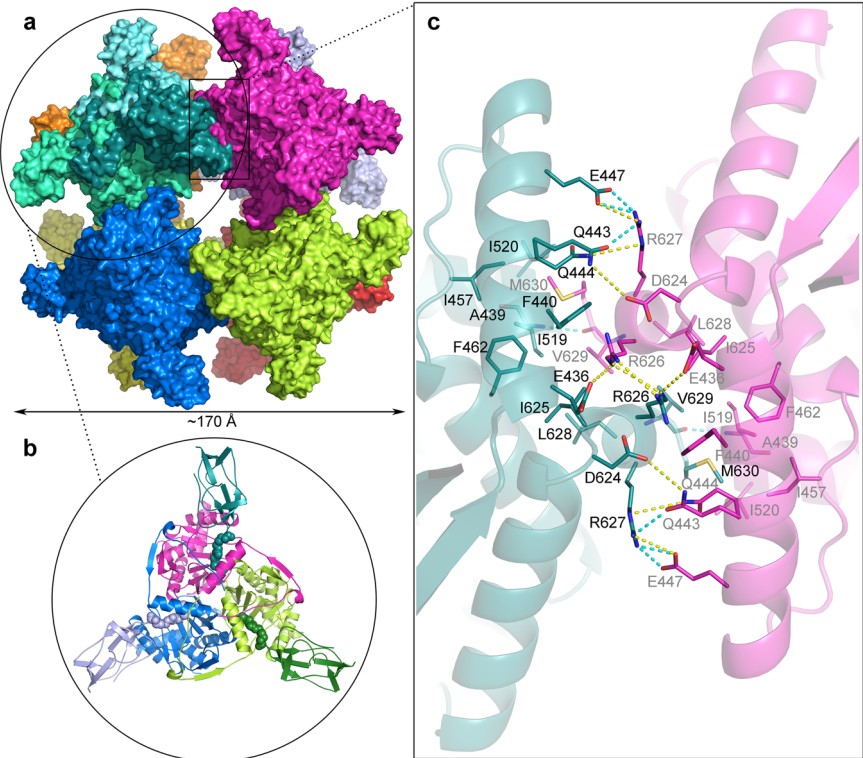

**Fig. 2 Structure of the 24-mer of *E. coli* dihydrolipoyl transacetylase (E2p). a** Surface representation of the overall structure of the E2p cubic core with each E2p trimer shown in a different color; E2p monomers in the top left E2p trimer are shown in different shades of cyan. **b** Cartoon structural representation of the E2p trimer. In the top E2p monomer the catalytic domain is colored magenta and the associated lipoyl domain in cyan with the dihydrolipoyllysine residue depicted as spheres, while in the other two E2p monomers the catalytic and lipoyl domains are colored in different shades of green and blue, respectively. **c** Molecular details of the contact interface between E2p trimers viewed along the two-fold axis. Interface residues located within the van der Waals distance are shown as sticks with residue numbers indicated for one E2p monomer (cyan) in black and for the second monomer (magenta) in gray. Pale cyan dashed lines represent hydrogen bonds and yellow dashed lines represent other polar interactions.

and we believe that in the context of the whole complex, given the high local concentration of covalently tethered LDs, the binding to E1p would be efficient even at the low affinity. We therefore assess that E1p might be occupied by the trapped LDs with oxidized lipoyllysine. Therefore, other LDs carrying oxidized lipoyllysine (the product of E3) would remain bound to E3 and thus block the E3 active site from the access of the substrate, which are the LDs with reduced dihydrolipoyllysine. These LDs with dihydrolipoyllysine could thus remain bound to the active site of E2p, as the product of E2p (Fig. 1a). Moreover, the reduced form explained the cryo-EM map better, so we modeled the prosthetic group as dihydrolipoamide, i.e., the product of the E2p-catalyzed reaction. The cryo-EM map did not suggest the presence of the S-acetyl modification on the dihydrolipoamide, in agreement with the mass-spectrometry data.

The structure of the trimeric assembly of the catalytic domain of *E. coli* E2p has been previously determined by X-ray crystallography (PDB code 4n72[37]), but the trimers did not assemble into the cubic core in the crystal. Our structure is highly similar to this crystal structure (RMSD of 0.7 Å for 245 aligned residues) and differs only in the short C-terminal helix, which represents the main contact region between the individual trimers, located at the 2-fold symmetry axis. It also differs slightly at the site of interaction with the lipoyl domain (residues 537–542 are slightly shifted outwards from the opening of the active channel). Interestingly, the structure of the N-terminal extended "elbows" in our native full-length E2p is essentially the same as in the truncated E2p crystal structure. Specifically, each "elbow" wraps around the surface of the neighboring E2p monomer and their N-termini

meet at the 3-fold axis with the terminal modeled residue being P384. Since the "elbow" is preceded in the sequence by the flexible linker connecting the catalytic domain to the peripheral subunit-binding domain, it has previously been speculated that it might be flexible in the native state and that the ordered conformation observed also in the crystal structure of E2p from *Azotobacter vinelandii* might be a crystallization artifact[48]. Our structural data provides evidence that this is indeed a native conformation of the enzyme, and we could even observe a map corresponding to the flexible linkers emanating from the center of the E2p trimer along the threefold axis towards the outer shell (Supplementary Fig. 4c).

**Interaction at the 2-fold axis.** The reconstruction of the native octameric assembly of the E2p catalytic core trimers allowed for the analysis of the trimer-trimer interface at the 2-fold axis. The interface area is 840 Å$^2$, includes 24 pairs of interfacing residues, and is stabilized mostly by hydrophobic interactions, but also by polar interactions including hydrogen bonds and salt bridges (Fig. 2c). The properties of the C-terminal residues are conserved in all E2p sequences, regardless of the size and symmetry of the quaternary assembly formed[49]. All the E2 trimer–trimer interfaces, including our *E. coli* E2p, are based on a conserved hydrophobic knob-into-hole interaction, where the short C-terminal helix represents a knob that fits into a hydrophobic hole in the neighboring E2 trimer[50]. Such knob-into-hole arrangement provides rotational flexibility that allows for the formation of different macromolecular assemblies, i.e., the octahedral 24-mer, the icosahedral 60-mer, or the 48-mer oblate

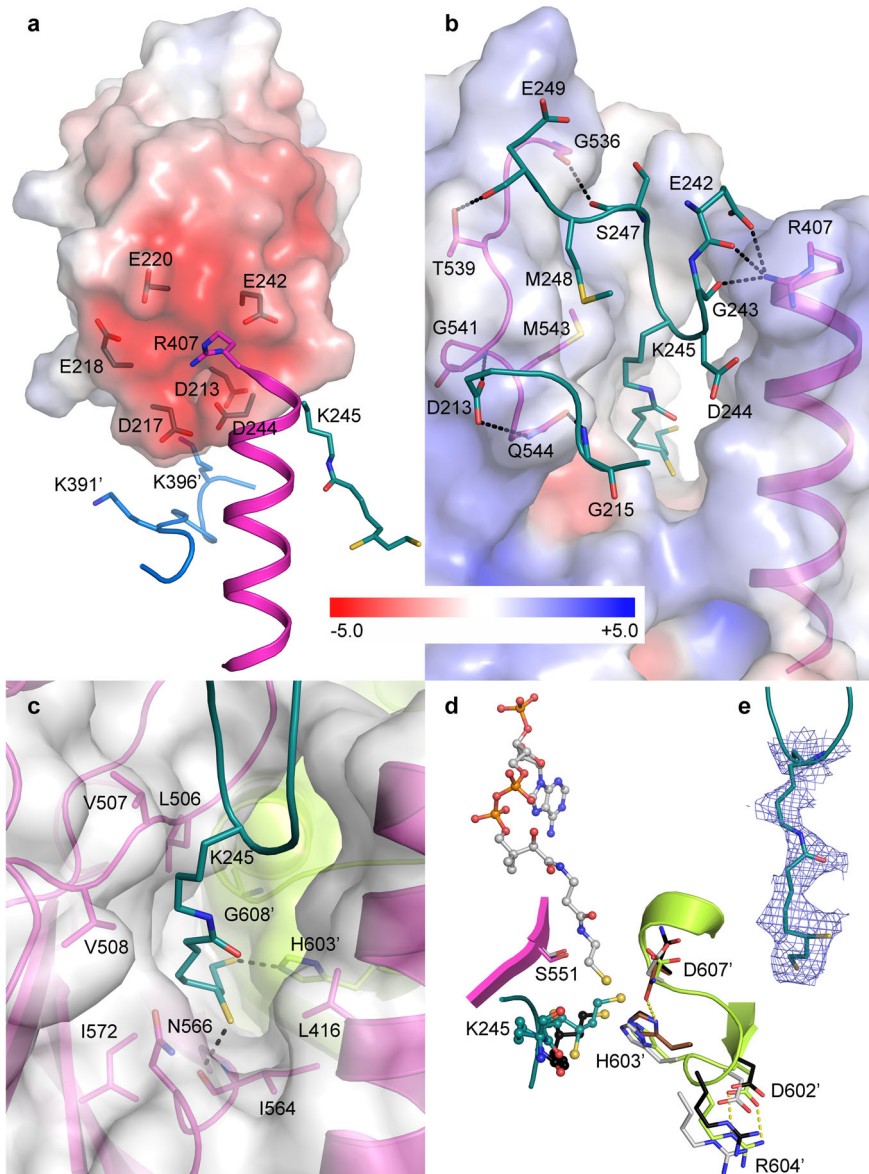

**Fig. 3 Interaction of the lipoyl domain with the catalytic domain of *E. coli* dihydrolipoyl transacetylase.** The color code for the individual domains corresponds to Fig. 2b. **a** Acidic patch on the surface of lipoyl domain interacts with the positive dipole and an arginine residue in the N-terminal tip of the catalytic domain helix H1, and with two basic residues from the N-terminal stretch of the neighboring catalytic domain. Residue D213 lies on the backside of the lipoyl domain. **b** Key interactions are visualized between the loops of the lipoyl and catalytic domains at the binding interface located around the entrance into the active site channel. In **a**, **b**, key residues are shown as sticks and surfaces of the lipoyl (**a**) and catalytic (**b**) domains are colored based on the electrostatic potential (values in kT/e). **c** Interaction of the dihydrolipoyllysine residue (cyan) with the residues in the active site channel at the interface of two catalytic domains (magenta and green, respectively). **d** Comparison of the active site of *E. coli* E2p (color code is the same as in **b**) with the structures of the complexes of *Azotobacter vinelandii* E2p with dihydrolipoate (PDB code 1eae, black) and oxidized CoA (PDB code 1ead, brown; only the catalytic His and Asn are shown)[70], and *Bos taurus* E2b with CoA (PDB code 2ii4, light gray)[72]. Residue numbers correspond to *E. coli* E2p. **e** Cryo-EM map for the dihydrolipoyllysine residue.

spheroid, according to the quasi-equivalence concept[50–55]. In addition, it also allows for the expansion and contraction of the E2 core, described as breathing for the *Saccharomyces cerevisiae* E2p core[56]. No such breathing was observed for *E. coli* E2p in this study or in a previous cryo-electron tomography study[40], which could be explained by an additional stabilization of this interface by the polar interaction of residues E447 and Q443 with R627, in contrast to the mainly hydrophobic interaction-based interface in the *S. cerevisiae* E2p interface[56]. Similar explanation was suggested also for *E. coli* E2o[56,57].

**Binding of lipoyl domain.** The lipoyl domain interacts with the catalytic domain through a striking electrostatic network. The lipoyl domain exhibits a large acidic patch that interacts with the positive dipole of the N-terminus of the catalytic domain helix H1 as well as with an arginine residue (R407) on the N-terminal tip of this helix (Fig. 3a). Moreover, the overall surface potential around the active site channel entrance in the catalytic domain is mostly electropositive (Fig. 3b). There are a number of possible electrostatic interactions between the two molecular surfaces: the key contact residue R407 can interact with both E242 and D244,

and possibly also with E218 and E220. Additionally, residues from the neighboring catalytic domain (marked with an apostrophe) can contribute to this interaction, namely K391' could interact with D217, or K396' with D213, all in a transient, dynamic manner (Fig. 3a). All these acidic residues are completely conserved in all three lipoyl domains of *E. coli* E2 (Supplementary Fig. 6a, sequences obtained from the Uniprot database[58]). We therefore believe that this electrostatic interaction likely functions as a rather nonspecific anchoring system for the two domains in the primary navigation of the S-acetyldihydrolipoamide moiety into the active channel, before the lipoyl domain establishes stable interactions with the catalytic domain. Specifically, two loops (residues 213–215 and 242–249) of the lipoyl domain form several hydrogen bonds with a loop of the catalytic domain (residues 536–544) located along the entrance into the active channel, and there is also a van der Waals interaction between methionine M248 from the lipoyl domain and methionine M543 from the catalytic domain (Fig. 3b). The dihydrolipoyllysine (K245) itself forms predominantly hydrophobic interactions with residues of the channel leading into the catalytic site, which is formed at the interface between two catalytic domains, and both thiol groups are engaged in polar interactions (Fig. 3c).

Electronegative surface patches are present in all E2 lipoyl domains with the three-dimensional structure determined (references[39,59–69] and PDB codes 2l5t, 2dnc, and 2dne) and although their exact location differs slightly between the individual proteins, they usually stretch over one face of the lipoyl domain starting from the lipoylated lysine towards the N-termini and C-termini located at the opposite end of the domain. Residues E242 and D244 are conserved in E2p lipoyl domains of all domains of life (Supplementary Fig. 6a). All E2-family enzymes with a determined three-dimensional structure (see refs. [45,48,50,52–54,57,70–76] and PDB codes 6h05, 3mae, 3l60) have an electropositive surface patch at helix H1. Basic residues are located at or near the tip of helix H1 (R407 in *E. coli*) in E2p enzymes from all domains of life (Supplementary Fig. 6b). This suggests that the non-specific electrostatic navigation mechanism can be common to all E2 systems. Similar electrostatic interaction between the catalytic and lipoyl domains has been suggested based on molecular modeling for the *E. coli* E2o[73]. Moreover, the crystal structure of the human E2p lipoyl domain in complex with pyruvate dehydrogenase kinase, where the binding plays a regulatory role for the PDHc, revealed an electrostatics-based interaction between the lipoyl domain (electronegative) and the kinase (electropositive)[63] (also present in other structures[64–66]). We therefore propose that the acidic patches in the lipoyl domains of E2-family enzymes might function as general binding sites, interacting with other binding partners than the E2 catalytic domains, too.

**Dihydrolipoyllysine in the active site**. The roughly 30 Å long active site channel is located at the interface between two catalytic domains and three such channels exist in each E2p trimer. S-acetyldihydrolipoamide and CoA bind at the two opposing sites of the channel and their reactive terminal sulfurs meet in the close proximity of the catalytic histidine (H603'), where the transacetylase reaction occurs. S-acetyldihydrolipoamide enters the active site from the outside of the cube and CoA from the large solvent-accessible cavity inside the hollow cube (the entrance into the inner cavity of the cube is about 30–40 Å wide). The catalytically active residues H603', D607', and S551[77,78] are located deep in the channel and H603' and D607' belong to the catalytic domain neighboring to the one which interacts with the lipoyl domain and forms the entrance for S-acetyldihydrolipoamide into the

active channel. The catalytic mechanism of *E. coli* E2p and other E2 enzymes is similar to that described before for chloramphenicol acyltransferase[49,79–82]. H603', stabilized by D607' in the correct tautomeric form, functions as a general base that removes a proton from the thiol group of CoA, which then attacks the carbonyl carbon of the acetyl moiety bound to dihyrolipoamide, forming a tetrahedral intermediate, which is stabilized by S551, and then falls apart resulting in the transfer of the acetyl group to CoA[53,70–72]. In our structure the reactive terminal sulfur (S8) of the dihydrolipoamide, which normally carries the acetyl group, is oriented towards a nitrogen atom (NE2) of the catalytic histidine H603' and is in a close proximity to S551 and D607' (Fig. 3d). The position of H603' in the active site is stabilized by a salt bridge formed between the neighboring residues D602' and R604', which is conserved in all the E2 proteins with their structures determined (refs. [48,50,53,54,57,70–76] and PDB codes 6h05, 3mae, 3l60) and the sequence D-H-R is conserved in all E2 sequences[49] (see also Supplementary Fig. 6b).

X-ray structures of E2 complexes with ligands were previously determined[70,72]. In particular, superposition of binary complexes of *A. vinelandii* E2p with dihydrolipoate[70] and *B. taurus* E2b with CoA[72] with our structure illustrates the position of the CoA-binding site and allows a comparison of ligand binding; we did not observe any cryo-EM map suggesting the presence of coenzyme A in our resting state data. The pose of the dihydrolipoyllysine residue in our structure is slightly different from that of the dihydrolipoate in the *A. vinelandii* E2p, which can be explained by the attachment to the lysine residue of the lipoyl domain in our structure together with the high conformational flexibility[21], and a lack of any specific polar interactions between the swinging arm and the channel. The conformation of the active site residues is also consistent with the homologous structures. *A. vinelandii* E2p has an asparagine residue in the position of aspartate D607', which is less frequently found in E2p sequences than aspartate (Supplementary Fig. 6b)[73]. This residue is expected to change its conformation upon ligand binding and form a long hydrogen bond with the catalytic histidine[70,71]. This conformation can be observed for example in the structure of *A. vinelandii* E2p in complex with oxidized CoA[70] (Fig. 3d), but not in the CoA-bound structure of *B. taurus* E2b[72], and is also not captured in our structure.

A substrate-mediated gating mechanism has been described for the *B. taurus* E2b, where the binding of CoA triggers a conformational change in the active site channel, which opens the otherwise closed gate in the channel and allows for the binding of S-acetyldihydrolipoamide at the other side of the channel[72]. The same mechanism likely applies to human E2p, where the CoA-induced gate opening also increases the affinity of lipoyl domains towards the E2p catalytic domains[45]. Since the prokaryotic E2 structures[37,57,70] all exhibit an open conformation in the absence of CoA, it is likely that the bacterial E2 enzymes do not use this gating mechanism, but rather depend on transcriptional control of the respective operons[72,83,84]. Indeed, our native cryo-EM structure suggests that the *E. coli* E2p exists in the resting state in the open-gate conformation and binds lipoyl-domain-bound dihydrolipoamide in the absence of coenzyme A.

In summary, we present here the structure of the *E. coli* E2p cubic core in a native resting state, comprising the octahedral assembly of the catalytic domains, which interact with lipoyl domains that carry a native dihydrolipoamide prosthetic group, bound in the active site of E2p. Our data provide structural information on the assembly of the dihydrolipoyl transacetylase catalytic core, including the interaction interface between the E2p trimers at the 2-fold axis. Most importantly, the structure reveals the interaction between an E2 enzyme catalytic domain and a lipoyl domain carrying a native lipoylation, and thereby shows

the molecular details of the binding of the native dihydrolipoyl-lysine into the active site of the enzyme.

## Methods

**Protein preparation.** The pyruvate dehydrogenase complex was isolated from *E. coli* K12 cells in an attempt to isolate the multienzyme complex synthetizing ubiquinone. The *E. coli* K12 cells containing a genome FLAG tag on the UbiF gene (GenScript, Piscataway, NJ, USA) were cultured in the LEX bioreactor (Epiphyte3 Inc., Toronto, Canada) for 24 h at 37 °C in the M9 medium with 0.5% succinate as the sole carbon source, yielding approx. 3 g of cells per 1 l of cell culture. Wild-type *E. coli* K12 cells were grown similarly in a small-scale 1 l culture in M9 medium with 0.5% succinate in a 5 l Erlenmeyer flask for 24 h at 37 °C while shaking at 180 rpm. The cells were disrupted by sonication in the lysis buffer (45 ml of lysis buffer per 10 g of cells), containing 20 mM Hepes pH 7.5, 150 mM NaCl, complete EDTA-free protease inhibitor cocktail (Roche, Basel, Switzerland) with a pinch of lysozyme and DNAse (both from Sigma Aldrich, St. Louis, MO, USA) and the lysate was ultracentrifuged (150,000×g, 60 min, 4 °C). The supernatant was incubated for 1.5 h at 4 °C tumbling with the anti-FLAG affinity gel (50 μl slurry per 10 ml of lysate, Sigma Aldrich, St. Louis, MO, USA) equilibrated in the lysis buffer. The resin was sedimented by centrifugation (3000×g, 3 min, 4 °C), washed twice with 100 resin volumes of purification buffer, and eluted in two steps with a total of 2 resin volumes of lysis buffer without protease inhibitors supplemented with the 3× FLAG peptide (Sigma Aldrich, St. Louis, MO, USA) as follows. In each elution step, the resin was tumbled for 30 min at 4 °C in a sealed spin column, followed by centrifugation of the spin column (3000×g, 3 min, 4 °C). The eluted protein was pooled and concentrated on 100 kDa MWCO spin concentrators (10,000×g, 4 °C) to approx. 3 mg/ml (assuming the absorbance at 280 nm = 1 for a 1 mg/ml solution), and immediately used for cryo-EM grid preparation, or flash frozen in liquid nitrogen and stored at −80 °C for further use in the activity assay, silver-stained SDS-PAGE, and mass spectrometry analysis. The yield was about 10 μg of the pyruvate dehydrogenase complex per 1 l of cell culture.

**Protein sample characterization.** Protein sample purity was analyzed by silver-stained SDS-PAGE on 4–12% Bis–Tris NuPAGE™ gel in NuPAGE™ MOPS SDS running buffer (200 V, 45 min; Thermo Fisher Scientific, Waltham, MA, USA). The molecular weight standard was PageRuler™ Plus Prestained Protein Ladder (4 μl; Thermo Fisher Scientific, Waltham, MA, USA) and 8 μg of the protein sample were loaded on the gel.

The identity of the protein complex was verified by mass-spectrometry analysis (LC-Orbitrap MS/MS) at the Mass Spectrometry Based Proteomics Facility, Uppsala University using standard protocols. Briefly, 10 μg of the frozen purified protein sample were reduced, alkylated, in-solution trypsin-digested, purified by ZipTip® (Sigma Aldrich, St. Louis, MO, USA) and dried. Dried peptides were resolved in 30 μl of 0.1% formic acid, 4× diluted, separated in reversed-phase on a C18-column, and electrosprayed online to a QExactive Plus Orbitrap mass spectrometer (Thermo Finnigan, Waltham, MA, USA) with 150 min gradient. Tandem mass spectrometry was performed applying HCD. Database searches were performed using the Sequest algorithm, embedded in Proteome Discoverer 1.4 (Thermo Fisher Scientific, Waltham, MA, USA) against the database of *Escherichia coli* strain K12 proteome extracted from Uniprot (release March 2020).

**Activity assay.** The pyruvate degydrogenase and α-ketoglutarate dehydrogenase complex activities were assayed based on a previously described method[11]. Briefly, 150 μl of assay buffer (5 mM pyruvate or α-ketoglutarate, 2.5 mM NAD, 0.2 mM thiamine pyrophosphate, 0.1 mM CoA, 0.3 mM dithiothreitol, 1 mM MgCl₂, 150 mM NaCl, and 20 mM Hepes-NaOH pH 7.5) were added to a microcuvette. The reaction was started after 1 min of following NADH production at 340 nm when 30 μl of the enzyme complex at the concentration of 0.3 mg/ml in the sample buffer (20 mM Hepes pH 7.5, 150 mM NaCl) were added. The reaction was monitored for 3 min. Specific activity was determined based on the protein concentration (absorbance at 280 nm = 1 for a 1 mg/ml solution) and an NADH extinction coefficient of 6.22 mM⁻¹ cm⁻¹. The specific activity is the average of two measurements.

**Western blot analysis.** For the immunoblot analysis, 2.5 μl of each fraction from the purification were boiled for 3 min in an equal volume of sample buffer (100 mM Tris·HCl pH 6.8, 4% SDS, 20% glycerol, 0.2% bromophenol blue, and 100 mM DTT added fresh) and run on a 16% polyacrylamide, 0.2% bisacrylamide gel (30 mA, 50 min) together with the PageRuler™ Prestained Protein Ladder, 10–180 kDa (Thermo Fisher Scientific, Waltham, MA, USA). Proteins were subsequently transferred to a nitrocellulose membrane (Amersham™ Protan® Premium Western blotting membranes, GE Healthcare, Chicago, IL, USA; 100 mA, 90 min) and treated with anti-lipoate antibody (a kind gift from Dr. Luke Szweda) diluted 1:500 in 5% skim milk powder (Sigma Aldrich, St. Louis, MO, USA) in TBS for 16 h at 4 °C. The membranes were washed in TBS before the secondary antibody (horseradish peroxidase-conjugated goat anti-rabbit antibody; cat no. #1705046, Bio-Rad, Hercules, CA, USA) diluted 1:10,000 in 5% skim milk powder (Sigma Aldrich, St. Louis, MO, USA) in TBS was applied (1 h, 23 °C). After another TBS wash, the membranes were developed using enhanced chemiluminiscence

substrate (WesternBright Quantum, Advansta, Menlo Park, CA, USA) and Fusion FX7 imaging system (Vilber, Collégien, France). Raw Western blot image and membrane preview are provided in the source data file.

**Mass spectrometry analysis of protein lipoylation.** Approx. 1 μg of purified PDHc was subjected to digestion using the SP3 method[85]. Acetonitrile was added to a final concentration of 50% and beads with captured proteins were washed twice with 70% ethanol. Ten microliters of trypsin/LysC digestion solution, containing 250 ng of trypsin (Sigma Aldrich, St. Louis, MO, USA) and 250 ng of LysC (Fujifilm Wako Chemicals USA, Richmond, VA, USA) in 100 mM Hepes, pH 8.5, were added and digestion was performed for 2 or 4 h. Generated peptides were washed twice with acetonitrile (200 μl) and peptides were eluted in 5% DMSO. The samples were acidified using formic acid to a final concentration of 2% and acetonitrile was added to a final concentration of 2%.

The instrumentation consisted of a nano LC 1200 chromatographic system coupled via a nano-spray ionization source to an Exploris 480 mass spectrometer (Thermo Fisher Scientific, Waltham, MA, USA). For peptide separation a 75 μm inner diameter in-house packed column (PoroShell 2.8 μm, Agilent Technologies, Santa Clara, CA, USA) was utilized and a binary solvent-based gradient (Thermo Fisher Scientific, Waltham, MA, USA) was applied as follows. The buffer B (80% acetonitrile in 0.1% formic acid) content was increased linearly from 5 to 30% within 70 min and further ramped to 45% within 10 min. Then, buffer B content was increased to 95% in a linear manner within 5 min and held there further for 5 min. Prior injection of the next sample, the column was re-equilibrated using 8 μl of buffer A (0.1% formic acid). MS1 spectra were acquired using a resolution of 60,000 at 200 *m/z*. The mass spectrometer operated in a data-dependent mode targeting the Top12 most intense peaks for quadrupole isolation and MS/MS spectra acquisition. The isolation window was set to 1.2 Th and the AGC target was defined as 500% and the resolution was set to 30,000 at 200 *m/z*. Dynamic exclusion was set to 20 s. The transfer capillary temperature was set to 275 °C and the Funnel RF level was set to 55.

**Mass-spectrometry data analysis.** The MaxQuant software with the implemented Andromeda search engine[86,87] were used for the analysis of acquired MS/MS spectra. The Fasta file for the protein (aceF gene from *E. coli* strain K12) was downloaded from the Uniprot database (accession code P06959, April 2021). The following lipoylation modification states were defined: oxidized form, reduced form, 1× acetylated form, 2× alkylated (carbamidomethylated) form, and also the potential 1× acetylated/1× alkylated form. The minimal Andromeda score for a modification was 25 and a localization probability above 0.9. The "modificationSpecificPeptide" and the lipoylation site tables were used for visualization of the results. InstantClue was utilized for visualization of the data[88].

**Cryo-EM sample preparation and data collection.** Quantifoil holey carbon grids (Quantifoil Micro Tools, Jena, Germany) were glow discharged (20 mA, 60 s, GloQube® Plus Glow Discharge System, Quorum Technologies, Laughton, UK) and 3 μl of protein sample diluted in 20 mM Hepes pH 7.5, 150 mM NaCl were applied onto each grid using the Vitrobot blotting robot (FEI, Hillsboro, OR, USA) at 4 °C and 100% humidity with 15 s of sample equilibration time before blotting for 3 s with the blot force set to 0. Grids were clipped, stored in liquid nitrogen, and screened on a Talos Arctica microscope operating at 200 kV and equipped with a Falcon III direct electron detector (FEI, Hillsboro, OR, USA) at the Stockholm node of the Swedish National Cryo-EM facility (SciLifeLab, Stockholm, Sweden). Three data sets were collected on a Titan Krios microscope (FEI, Hillsboro, OR, USA) operating at 300 kV and equipped with a K2 BioQuantum 4k × 4k direct electron detector (Gatan, Pleasaston, CA, USA) at the Umeå node of the Swedish National Cryo-EM facility (Umeå University, Umeå, Sweden), using similar data collection parameters (see Supplementary Table 2) and the EPU software version 2.7. For dataset 1, a Quantifoil R2/2 Cu 300 mesh holey carbon grid was used and the protein concentration was 0.6 mg/ml. For dataset 2, a Quantifoil R1.2/1.3 Cu 300 mesh holey carbon grid was used and the protein concentration was 1.4 mg/ml. For dataset 3, a Quantifoil R2/2 Cu 300 mesh holey carbon grid pre-coated with a 0.2 mg/ml suspension of graphene oxide (Sigma-Aldrich, St. Louis, MO, USA) after glow discharge (40 mA, 90 s, GloQube® Plus Glow Discharge System, Quorum Technologies, Laughton, UK) and 2.5 μl of the protein sample at the concentration of 0.7 mg/ml were used.

**Cryo-EM data processing and model building.** All the data processing was carried out in cryoSPARC version 2.15[89,90] (Supplementary Figs. 4 and 5). A total of 20,133 movies were recorded, 3,708 in dataset 1, 3,558 in dataset 2, and 12,867 in dataset 3. A total of 799,790 particles were automatically template-picked from a total of 20,078 movies (the box size was 300 px), and after four rounds of particle filtering using 2D classification, 42,266 particles were selected for 3D classification and refinement. Heterogeneous 3D refinement (3D classification) was performed, where 12,832 particles were excluded, because they were lacking the lipoyl domains. The remaining 29,434 particles were included in the final 3D volume refinement with imposed O symmetry, which yielded the final map at 3.16 Å resolution, calculated based on the gold-standard Fourier shell correlation (FSC) of 0.143[91] (Supplementary Fig. 5). The *E. coli* pyruvate dehydrogenase complex E2p

core model was built into the map using a combination of automated docking of the crystal structure of the *E. coli* E2p catalytic domain (PDB 4n72[37]), and the NMR solution structure of the *E. coli* E2p innermost lipoyl domain (PDB 1qjo[39]) using Phenix version 1.16[92] and manual building in Coot version 0.8.9.1[93] together with real space refinement in Phenix and CCPEM version 1.4.1[94]. The cryo-EM data collection, refinement, and validation statistics are listed in Supplementary Table 2. The model coordinates and cryo-EM maps were deposited to the Protein Data Bank (accession code 7b9k) and to the Electron Microscopy Data Bank (accession code EMD-12104).

For an additional analysis of the whole pyruvate dehydrogenase complex (Supplementary Fig. 4), a box size of 600 px was used for particle extraction, followed by particle down-sampling to a box size of 300 px. A total of 28,674 down-sampled particles were included in the 3D classification and refinement without any symmetry imposed. For the 2D classification of the outer shell of the pyruvate dehydrogenase complex, a mask covering the E2p core was used for particle subtraction and the subtracted particles (28,674) were 2D classified to visualize the E1p and/or E3 components.

The quality of the final E2p model was evaluated using Phenix version 1.16[92] and the MolProbity server[95], sequence alignments were made using the Clustal Omega server[96], the Dali[97] and PISA[98] servers were used for structure analysis, and all structure and volume representation figures were created in the PyMOL Molecular Graphics System version 2.2.3 (Schrödinger, LLC) and UCSF Chimera version 1.13.1[99], respectively.

**Reporting summary**. Further information on research design is available in the Nature Research Reporting Summary linked to this article.

## Data availability
The cryo-EM reconstruction has been deposited in the Electron Microscopy Data Bank under the EMDB accession code EMD-12104 and the atomic model has been deposited in the Protein Data Bank under the PDB accession code 7b9k. Source data are provided with this paper.

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

## Acknowledgements

We would like to thank Marta Carroni, Dustin Morado, Julian Conrad, Karin Wallden, and Jonathan Davies for advice and discussion about cryo-EM data processing and reporting. We thank the Swedish National cryo-EM facility staff Michael Hall, Thomas Heidler, Julian Conrad, and Marta Carroni for their help during the acquisition of the cryo-EM data, Alexander Manoilov and Ganna Shevchenko from the Mass Spectrometry Based Proteomics Facility, Uppsala University, for the mass-spectrometry analysis, and Zdenek Kukacka for mass-spectrometry discussions. We also wish to thank Katharina Stephan for her advice on the purification, Markel Martinez-Carranza and Agnes Moe for their assistance with EM, and to the team of Prof. Martin Ott for advice and discussions. Cryo-EM sample screening, optimization, and data collection were performed at the Cryo-EM Swedish National Facility, funded by the Knut and Alice Wallenberg, Family Erling Persson and Kempe Foundations, SciLifeLab, Stockholm University and Umeå University, at the SciLifeLab node in Stockholm and the Umeå Core Facility for Electron Microscopy node. This work was supported by the Swedish Research Council (grant 2018-03406 to P.S.) and the Swedish Cancer Society (grant to P.S.). The work of J.S. was supported from ERDF/ESF, OP RDE, project "IOCB Mobility" (No. CZ.02.2.69/ 0.0/0.0/16_027/0008477) granted to the Institute of Organic Chemistry and Biochemistry of the Czech Academy of Sciences.

## Author contributions

J.S. planned and performed experiments, analyzed the data, and wrote the manuscript draft with input from J.B. and H.N., J.B. and H.N. performed experiments and analyzed the data, M.O. participated in supervision, and P.S. conceived and supervised the study, analyzed the data, and revised the manuscript. All authors commented on the final manuscript.

## Funding

## Competing interests

The authors declare no competing interests.
