## [Peer Review File · Nature Communications]

Structure of the native pyruvate dehydrogenase complex reveals the mechanism of substrate insertionREVIEWER COMMENTS

Reviewer #1 (Remarks to the Author):

This manuscript by Skerlova et al reports the cryo-EM structure of a native *E. coli* pyruvate dehydrogenase complex (PDHc). The cryo-EM data analysis provides a low-resolution reconstruction of the whole complex and a near-atomic resolution structure of the cubic core complex of the catalytic and lipoyl domains of E2p. It is interesting to visualize the binding of the lipoyl domain to the catalytic domain and the lipoyllysine moiety inserting into the substrate entrance of the catalytic domain in the native PDHc. The authors compared their structure of *E. coli* E2p core complex with a number of structures of the core or domains of E2 from the previous studies, revealing their high similarities. The manuscript goes further to describe the interactions at the 2-fold interface of the E2p core, the binding of the lipoyl domain to the catalytic domain, and the interactions of the lipoyllysine moiety in the active site. It is of significance to study the structure of the native PDHc, but unfortunately this manuscript does not bring sufficient new insights into the structure, mechanism, or activity regulation of the PDHc. Most of the messages carried in this manuscript have been demonstrated or suggested in the previous studies. It is recommended that the authors perform further analyses and comparison of their structure and additional investigations besides the structural study to reveal remarkable new knowledge about the PDHc, a central metabolic gatekeeper that is associated with numerous human diseases.

Comments:

- 1) This manuscript does not discuss why they ended up with getting PDHc in an attempt to purify another protein UbiF that carries the affinity FLAG tag. It makes the protein purification result more confusing that the mass spectrometry analysis (Supplementary Table 1) did not detect UbiF. It is understandable that sometimes co-purification of other protein(s) happens, but it is worth carefully investigating why/how this happened. I am concerned that this PDHc sample from the off-target purification may not represent the general population of PDHc (in the claimed resting state) in *E. coli*, if the protein purification strategy is not clearly established.
- 2) The authors assumed that the cellular pyruvate level was low and therefore the PDHc was in the resting state when succinate was used as the sole carbon source to support *E. coli* growth. They need to provide either data or literature to support this assumption. It is known that pyruvate is produced not only from glycolysis but also from other sources, such as oxidative decarboxylation of malate which is an intermediate after succinate in the Krebs cycle. Therefore, the availability of pyruvate may not be low enough to keep the PDHc in the resting state.
- 3) The identification of the lipoyllysine moiety as its reduced form is speculative. Did the mass spectrometry analysis detect peptides containing lipoyllysine? These peptides may tell the state(s) of the lipoyllysine. Another suggestion is to add certain substrates to trap the PDHc in a defined state before the cryo-EM study.

Reviewer #2 (Remarks to the Author):

I appreciate the authors' efforts in helping advance an important area of biological and medical relevance. There are a number of comments and questions that I feel, if addressed, would lead to a stronger and more informative paper, and these are now described.

- 1) I would strongly suggest a minor title change. Since no substrate or analog was actually present or included in the results, a more informative and precise title could be obtained by replacement of the words "substrate insertion" with "lipoyl domain-E2 active site binding" or something to that effect, perhaps providing the full protein name instead of "E2" since the general reader may not know what E2 is. For example, Pg 1, doesn't really tell how LD inserts substrate into the active site, since no substrate was present.
- 2) There is very little information provided regarding sample preparation, purification, etc. I think many readers would appreciate providing such information.

Pg 3, The authors may also want to point out that the thiamine based cofactor is often referred to as TDP or ThDP (for thiamine diphosphate, vitamin B1 diphosphate), in addition to the older nomenclature, thiamine pyrophosphate, TPP).

The authors should point out that E3 is also FAD dependent.

Line 66, the authors should also include "symmetry" as varying.

Pg 4, line 78, the core was thought to be truly cubic, and it's the entire complex that was thought to be octahedral (because of the E3 DIMERIC components), and 432 symmetry is by definition, consistent with a cube.

Line 84, I wouldn't use the word "alternate" in delivering substrates, as it implies some type of synchronization. I am not aware of any evidence for this, and it's an entirely open question.

Line 90. I'm not sure it's appropriate to refer to dihydrolipoalysine as the "resting" state of the enzyme, as in the "dihydro" state the acetyl group would already have been transferred to CoA. If that were truly the "resting" state there would be no need for E3, as an "open" (i.e "dihydro") dithiolane ring on lipoamide would be able to repeatedly pick up acetyl groups on E1, and transfer them to CoA. This should be corrected or at least explained.

In a related issue, I think part of Fig 1 is either incorrect, or at best misleading. In the figure, I presume arrows are representing a progression between states, and if so that indicates LD bound to an E1 active site would move directly to bind to an E3 active site. This would be totally unproductive, if even possible, as the state of the lipoamide exiting E1 (open ring, one S acetylated) is NOT the proper substrate for E3 (open ring, no acetylation, hence DIHYDRO). After exiting E1, the acetylated lipoamide must enter an E2 active site, transfer the acetyl group to CoA, and it's the newly created product, dihydrolipoamide WITHOUT any acetyl groups that's the substrate for E3.

Pg 5, Line 104, and elsewhere throughout paper, I don't believe "electron density" map is the appropriate name for a cryo-EM map, as it's some type of "potential" map, although they often look similar to crystallographic electron density maps, at least for non-charged residues.

Response to reviewers

We thank the reviewers' for their positive response and constructive suggestions. We believe the resulting changes have improved our manuscript.

Reviewer comments

Reviewer #1 (Remarks to the Author):

This manuscript by Skerlova et al reports the cryo-EM structure of a native *E. coli* pyruvate dehydrogenase complex (PDHc). The cryo-EM data analysis provides a low-resolution reconstruction of the whole complex and a near-atomic resolution structure of the cubic core complex of the catalytic and lipoyl domains of E2p. It is interesting to visualize the binding of the lipoyl domain to the catalytic domain and the lipoyllysine moiety inserting into the substrate entrance of the catalytic domain in the native PDHc. The authors compared their structure of *E. coli* E2p core complex with a number of structures of the core or domains of E2 from the previous studies, revealing their high similarities. The manuscript goes further to describe the interactions at the 2-fold interface of the E2p core, the binding of the lipoyl domain to the catalytic domain, and the interactions of the lipoyllysine moiety in the active site. It is of significance to study the structure of the native PDHc, but unfortunately this manuscript does not bring sufficient new insights into the structure, mechanism, or activity regulation of the PDHc. Most of the messages carried in this manuscript have been demonstrated or suggested in the previous studies. It is recommended that the authors perform further analyses and comparison of their structure and additional investigations besides the structural study to reveal remarkable new knowledge about the PDHc, a central metabolic gatekeeper that is associated with numerous human diseases.

Comments:

1) This manuscript does not discuss why they ended up with getting PDHc in an attempt to purify another protein UbiF that carries the affinity FLAG tag. It makes the protein purification result more confusing that the mass spectrometry analysis (Supplementary Table 1) did not detect UbiF. It is understandable that sometimes co-purification of other protein(s) happens, but it is worth carefully investigating why/how this happened. I am concerned that this PDHc sample from the off-target purification may not represent the general population of PDHc (in the claimed resting state) in *E. coli*, if the protein purification strategy is not clearly established.

Thank you for pointing out the absence of UbiF in Supplementary Table 1. In fact, the mass spectrometry analysis did detect low amounts of UbiF together with all the other components of the ubiquinone synthesizing metabolon. The higher amount of PDHc relative to the Ubi complex in the purified sample reflects the relative amounts of the PDHc and Ubi complex in the cell, which needs higher amounts of the PDHc for providing energy than of the Ubi

complex for the synthesis of ubiquinone. We have updated Supplementary Table 1 and now it includes data about the Ubi complex components as well.

The purification protocol that we established, as described in the methods section, repeatedly yielded pure PDHc, so it was not a single serendipitous purification.

*We have noticed that another research team who isolated the ubiquinone synthesizing metabolon using a two-step affinity purification protocol employing the SPA-tag (3xFLAG-tag combined with the CBP domain) also identified by mass spectrometry the components of the PDHc as impurities in their sample. The amount of the PDHc was lower than in our case probably due to the use of three FLAG tags instead of one and due to the second purification step using the CBP domain. Reference: Hajj Chehade M, Pelosi L, Fyfe CD, Loiseau L, Rascalou B, Brugière S, Kazemzadeh K, Vo CD, Ciccone L, Aussel L, Couté Y, Fontecave M, Barras F, Lombard M, Pierrel F. A Soluble Metabolon Synthesizes the Isoprenoid Lipid Ubiquinone. *Cell Chem Biol.* 2019 Apr 18;26(4):482-492.e7. doi: 10.1016/j.chembiol.2018.12.001. Epub 2019 Jan 24. PMID: 30686758. (For the MS analysis, see supplementary information.)*

We hypothesize that the off-target purification of the PDHc might have happened due to the presence of the sequence motif DDYR (residues 819-822) in PDHc E1, which slightly resembles a part of the FLAG-tag sequence (DYKDDDDK). This sequence motif is not conserved in the homologous 2-oxoglutarate dehydrogenase E1 and despite the presence of low amounts of the 2-oxoglutarate dehydrogenase complex components in the sample detected by mass spectrometry, the 2-oxoglutarate dehydrogenase activity was not observed in our purified PDHc sample, so the anti-FLAG resin has relatively selectively captured the PDHc.

*To test this hypothesis, we have performed a purification following the same purification protocol using two *E. coli* strains in parallel: the *E. coli* K12 cells with the genome FLAG-tag on UbiF and also the wild type *E. coli* K12 cells without any FLAG epitope in the genome. The purification yielded similar amounts of PDHc from both strains, and was therefore not dependent on the presence of the FLAG-tag. New Supplementary Fig. 2 was added to the Supplementary material and the description of the wild type *E. coli* K12 cultivation, western blot analysis, and specific activity determination was added to the Methods section.*

The following text was added to the first paragraph of the Results and Discussion section:

*"The *E. coli* K12 cells had a FLAG tag in the genome on the gene coding for the UbiF protein, as they were originally designed for isolation of the ubiquinone synthesizing complex, which was also present in low amounts in the purified PDHc sample (Supplementary Table 1). The isolation of the PDHc using the anti-FLAG resin was reproducible and independent of the presence of the FLAG epitope, as similar purification outcome was achieved with wild type *E. coli* K12 cells (Supplementary Fig. 2)."*

2) The authors assumed that the cellular pyruvate level was low and therefore the PDHc was in the resting state when succinate was used as the sole carbon source to support *E. coli* growth. They need to provide either data or literature to support this assumption. It is known that pyruvate is produced not only from glycolysis but also from other sources, such as oxidative decarboxylation of malate which is an intermediate after succinate in the Krebs cycle. Therefore, the availability of pyruvate may not be low enough to keep the PDHc in the resting state.

We are aware of the presence of pyruvate in the cell originating from other processes such as transamination of alanine, oxidative decarboxylation of malate, or from oxaloacetate through phosphoenolpyruvate. We think that these biochemical pathways are not favored over the citric acid cycle progression from succinate, and that the cellular levels of pyruvate are therefore significantly lower than in the state when pyruvate is produced from glycolysis.

In response to this concern, we have reanalyzed the mass spectrometry data on the PDHc sample to identify the state of the lipoyl modification (see also the response to question no. 3). The data indicate that a majority of the identified peptides carried the lipoate in a non-acetylated form (oxidized, reduced, or 2x carbamidomethylated) and the acetylated form (the acetylated form only; the potential acetylated and 1x carbamidomethylated form was not detected) was present only in a small amount. The MS data clearly support the assumption of a low pyruvate level in the cells and the presence of the PDHc in the resting state. A new Supplementary Fig. 3 was added together with a method description in the Methods section. The complete mass spectrometry data can be found in the full version of Supplementary Table 1 provided as an Excel file. The description of the mass-spectrometry results was added to pages 8-9.

3) The identification of the lipoyllysine moiety as its reduced form is speculative. Did the mass spectrometry analysis detect peptides containing lipoyllysine? These peptides may tell the state(s) of the lipoyllysine. Another suggestion is to add certain substrates to trap the PDHc in a defined state before the cryo-EM study.

We have confirmed the presence of the lipoyl modification in the sample by an anti-lipoate antibody. See Supplementary Fig. 2b. This information was now added to the first paragraph of Results and discussion, page 5.

In order to determine the state of the lipoyllysine modification, we established collaboration with mass-spectrometry experts from the Max-Planck-Institute for Biology of Ageing. In this collaboration, we reanalyzed the data obtained from the original mass spectrometry analysis specifically aiming for the identification of lipoyllysine modifications in oxidized, reduced, and acetylated forms and also the 2x carbamidomethylated form and the potential acetylated and 1x carbamidomethylated form (after the reduction and alkylation). We detected the lipoylation on residues K41, K144, and K245. This information

was now added to the first paragraph of Results and discussion, page 5. The complete mass spectrometry data are in the newly provided full version of Supplementary Table 1 (Excel file) and the main results are presented in the new Supplementary Fig. 3. Oxidized lipoyllysine was the most abundant form of the cofactor, but reduced and acetylated forms of lipoate were also present (see the new Supplementary Fig. 3a). Unfortunately, sample reduction and alkylation were performed in order to achieve efficient protein digestion and good coverage.

In order to eliminate the effect of the reducing agent on the oxidation state of the lipoylation, we analyzed the purified PDHc sample using trypsin-based protein digestion without the addition of reducing and alkylating agents. We observed that the digestion efficiency was significantly reduced and fewer peptides were detected. Therefore, we optimized several experimental parameters (gradient settings, digestion time) and we were able to detect the lipoylation on residues K41, K141, and K245 (Supplementary Table 1), but only in the oxidized form. The other states of the cofactor could not be detected at all, not even the stable acetylated form present in the original MS data from the reduced and alkylated sample, indicating reduced coverage of modified peptides.

We also think that the reduced form of lipoate might only be stable when protected inside the protein's active pocket. During sample preparation for mass spectrometry, the protein is denatured in SDS and the lipoate gets exposed and, since the two thiol moieties are spatially close to each other, might easily become spontaneously oxidized in the surrounding oxidative environment. As a result, we could not unambiguously determine the oxidation state of the lipoate in the sample.

We expect that the PDHc in the resting state contains a mixture of oxidized and reduced lipoyllysines and even if the oxidized form prevails, the lipoyl domains captured bound to the E2 catalytic core are those that carry the reduced form of lipoyllysine. The reduced form is the product of the E2 active site and its binding to the active site makes better biological sense than binding of the oxidized form of lipoate that is neither a substrate, nor a product of the E2 active site and, with respect to the overall catalytic cycle of the PDHc, should not have a high affinity towards the E2 active site. Moreover, the shape of the cryo-EM map in the E2 active site corresponds better to the reduced form of lipoate than to the closed-ring oxidized form. Considering all the above points, we have decided to model the lipoate in the reduced form as dihydrolipoyllysine.

This reasoning is described in the manuscript text on pages 8 and 9, together with the reference to the new mass spectrometry data.

Regarding the suggestion to add certain substrates to the complex before the cryo-EM experiments, we believe that the key to the successful capture of the lipoyl domains bound to the E2 catalytic core was in fact trapping the complex in a resting state. The potential substrates to be added would be pyruvate and coenzyme A, which would keep the PDHc in an active state.

In an attempt to study the complex in the active state, we performed cryo-EM analysis on a PDHc sample isolated from E. coli cells cultivated in the presence of glucose. This cryo-EM data showed a majority of particles lacking well-resolved LDs bound to the E2 catalytic cores and we did not achieve high-resolution reconstruction. We speculate that these PDHc particles in which the LDs are dynamic and not bound to the E2 catalytic core might represent the active state of the complex.

Reviewer #2 (Remarks to the Author):

I appreciate the authors' efforts in helping advance an important area of biological and medical relevance. There are a number of comments and questions that I feel, if addressed, would lead to a stronger and more informative paper, and these are now described.

1) I would strongly suggest a minor title change. Since no substrate or analog was actually present or included in the results, a more informative and precise title could be obtained by replacement of the words "substrate insertion" with "lipoyl domain-E2 active site binding" or something to that effect, perhaps providing the full protein name instead of "E2" since the general reader may not know what E2 is. For example, Pg 1, doesn't really tell how LD inserts substrate into the active site, since no substrate was present.

We have changed the title to: "Structure of the native pyruvate dehydrogenase complex reveals the mechanism of dihydrolipoyllysine insertion"

2) There is very little information provided regarding sample preparation, purification, etc. I think many readers would appreciate providing such information.

We have described the complete purification procedure in the original manuscript and no purification or sample preparation steps are missing in the description. The purification protocol is indeed this simple and also reproducible. Using this protocol, we have repeatedly obtained a PDHc complex sample in an amount and purity sufficient for cryo-EM experiments.

Pg 3, The authors may also want to point out that the thiamine based cofactor is often referred to as TDP or ThDP (for thiamine diphosphate, vitamin B1 diphosphate), in addition to the older nomenclature, thiamine pyrophosphate, TPP).

We have pointed this out as suggested: "...thiamine pyrophosphate (TPP, also often referred to as TDP or ThDP for thiamine/vitamin B1 diphosphate)..."

The authors should point out that E3 is also FAD dependent.
We have pointed this out as suggested: "...by flavin adenine dinucleotide (FAD)-dependent E3..."

Line 66, the authors should also include "symmetry" as varying.

The symmetry was included as varying, as suggested: "The size, composition, and symmetry of the oxoacid dehydrogenase complexes vary in different organisms,..."

Pg 4, line 78, the core was thought to be truly cubic, and it's the entire complex that was thought to be octahedral (because of the E3 DIMERIC components), and 432 symmetry is by definition, consistent with a cube.

Thank you for this comment. The formulation of the sentence was changed to: "In Gram-negative bacteria the E2p catalytic domains are arranged in eight tight trimers further associated into a 24-meric cubic structure with an E2p trimer in each corner of the cube with a 432 symmetry^{6,17,18}."

Line 84, I wouldn't use the word "alternate" in delivering substrates, as it implies some type of synchronization. I am not aware of any evidence for this, and it's an entirely open question.

The word "alternate" was replaced with "participate".

Line 90. I'm not sure it's appropriate to refer to dihydrolipoyllysine as the "resting" state of the enzyme, as in the "dihydro" state the acetyl group would already have been transferred to CoA. If that were truly the "resting" state there would be no need for E3, as an "open" (i.e "dihydro") dithiolane ring on lipoamide would be able to repeatedly pick up acetyl groups on E1, and transfer them to CoA. This should be corrected or at least explained.

We believe that the reduced dihydrolipoyllysine is only present in a fraction of all LDs, while the rest is indeed present in the oxidized state. Hence, we do not refer to only the dihydrolipoyllysine as the resting state, but rather to the situation in the whole complex – a mixture of oxidized and reduced forms of lipoyllysine. Since the (dihydro)lipoyllysine moiety is rather hydrophobic, we assume that it prefers to exist buried in an active site. Reduced dihydrolipoyllysine can theoretically bind to E2 and E3, while oxidized lipoyllysine can bind to E3 and E1. Due to the lack of pyruvate in the cells, E1 is not active and is saturated by the trapped LDs with oxidized lipoyllysine. Therefore, other LDs carrying oxidized lipoyllysine as the product of E3 remain bound to E3 and thus block the E3 active site from the access of the substrate, which are the LDs with reduced dihydrolipoyllysine. These LDs with reduced dihydrolipoyllysine thus remain bound to the active site of E2 (as the product of E2). We call this situation a resting state, because the whole complex is not performing its catalytic activity due to the lack of pyruvate.

The corresponding text on pages 8 and 9 was rewritten to explain more clearly our rationale behind the resting state.

In a related issue, I think part of Fig 1 is either incorrect, or at best misleading. In the figure, I presume arrows are representing a progression between states, and if so that indicates LD bound to an E1 active site would move directly to bind to an E3 active site. This would be totally unproductive, if even possible, as the state of the lipoamide exiting E1 (open ring, one S acetylated) is NOT the proper substrate for E3 (open ring, no acetylation,

hence DIHYDRO). After exiting E1, the acetylated lipoamide must enter an E2 active site, transfer the acetyl group to CoA, and it's the newly created product, dihydrolipoamide WITHOUT any acetyl groups that's the substrate for E3.

Thank you for noticing this mistake. Figure 1f is indeed incorrect; we have by mistake swapped the last two reaction steps. This was corrected in the figure and the LD now binds first to the active site of E2 and then to the active site of E3 in the corrected scheme.

Pg 5, Line 104, and elsewhere throughout paper, I don't believe "electron density" map is the appropriate name for a cryo-EM map, as it's some type of "potential" map, although they often look similar to crystallographic electron density maps, at least for non-charged residues.

Thank you for pointing this out. We have replaced the term "electron density" by "reconstruction", "cryo-EM map", or just "map" throughout the entire manuscript and supplementary material.

Further changes to the manuscript: Hendrik Nolte was added into the author list, Author contributions were updated and Acknowledgements were edited.

REVIEWER COMMENTS

Reviewer #1 (Remarks to the Author):

The authors did an excellent job to address the reviewer's comments.

Reviewer #2 (Remarks to the Author):

This is an improved manuscript that should be useful to others, by shedding light on a significant biological system critical to energy production. There are a few remaining issues I think, if addressed, would give rise to a stronger paper.

Title- I would change "mechanism of dihydrolipoamide insertion" to "dihydrolipoamide interactions within the complex core"

This more precisely describes the paper, and draws focus to the component that's featured.

Line 28, delete "to insert substrate into" and add "at" the active site. I would NOT use the word substrate, here and elsewhere, as there was ambiguity in the nature of the observed, bound species. What was seen and modeled was apparently dihydrolipoamide-E2 (that's what is evident in Fig 3 panels). This is a "PRODUCT", along with acetyl-CoA, of the E2 reaction and NOT a substrate for it! It's a substrate for E3, while only S-acetyldihydrolipoamide (and CoA) are substrates for E2!

Note! This paper will convey very useful information, and I think it's important to be formally correct in all aspects, especially when denoting substrates and products of its various enzymatic components and reaction steps.

Line 71. I would change "the dihydro..." to "a dihydro...". This may seem picky, but it's not always the same E2 when different substrates are involved. It's only E3 that is identical in the various complexes within a species.

Line 84, I would change to "can carry" from "carries", as the LD domains are not always fully lipoylated.

Again, this may seem picky, but let's be precise.

Line 92, I would add "(in the absence of substrates)" after "resting state" to clarify.

Lines 96, I would change "inserts" to "could insert" or "might insert", or even "likely inserts" as you didn't see a substrate here; you may have seen a PRODUCT presumably ready to be released, as the nature of the lipoamide state was not clear. Also, it should be mentioned that the "other" substrate (CoA) is not present in this study, although a model was utilized from a related structure to provide insight. That's certainly appropriate in this case, but should be made clear early on.

Lines 163 and 164, I would change "of the E2p core" to "of the entire complex." It's too confusing with both the words "octahedron" and "cubic" used in the same sentence to describe the core, and it's only the core thought to be cubic.

Lines 192 & 193, It's well known that in the absence of substrate (pyruvate) E1 does NOT bind LD, regardless of the lipoamide state. I think there may be serious errors in the description/assessment of various lipoamide states abilities to bind to the components in this and the next few lines. If I'm mistaken, references for these assessments are needed here as they are not obvious or common knowledge, even to those familiar with the field.

We thank the reviewers for their constructive suggestions and answer them below in a point-by-point manner.

Reviewer #1 (Remarks to the Author):

The authors did an excellent job to address the reviewer's comments.

Reviewer #2 (Remarks to the Author):

This is an improved manuscript that should be useful to others, by shedding light on a significant biological system critical to energy production. There are a few remaining issues I think, if addressed, would give rise to a stronger paper.

Title- I would change “mechanism of dihydrolipoyllusine insertion” to “dihydrolipoyllisine interactions within the complex core”

This more precisely describes the paper, and draws focus to the component that's featured.

We agree with the reviewer that the formulation “mechanism of dihydrolipoyllisine insertion” is not suitable, as it is not the reduced form of lipoyllisine that is inserted into the E2 active site. We think though that the “dihydrolipoyllisine interactions within the complex core” does not fully describe the general findings of the paper, but only one structural aspect. We would prefer to replace the word “dihydrolipoyllisine” by the word “substrate” in the title, as it is also the structural information on the interaction of the LD with the E2 core that explains how the substrate is delivered into the active site. We also believe that this title will attract more readers to the manuscript.

Line 28, delete “to insert substrate into” and add “at” the active site. I would NOT use the word substrate, here and elsewhere, as there was ambiguity in the nature of the observed, bound species. What was seen and modeled was apparently dihydrolipoamide-E2 (that's what is evident in Fig 3 panels). This is a “PRODUCT”, along with acetyl-CoA, of the E2 reaction and NOT a substrate for it! It's a substrate for E3, while only S-acetyldihydrolipoamide (and CoA) are substrates for E2!

Note! This paper will convey very useful information, and I think it's important to be formally correct in all aspects, especially when denoting substrates and products of its various enzymatic components and reaction steps.

We have changed the text as requested.

Line 71. I would change “the dihydro...” to “a dihydro...”. This may seem picky, but it's not always the same E2 when different substrates are involved. It's only E3 that is identical in the various complexes within a species.

We have changed the text as requested.

Line 84, I would change to “can carry” from “carries”, as the LD domains are not always fully lipoylated.

Again, this may seem picky, but let's be precise.

We have changed the text as requested.

Line 92, I would add “(in the absence of substrates)” after “resting state” to clarify.

We have changed the text as requested.

Lines 96, I would change “inserts” to “could insert” or “might insert”, or even “likely inserts” as you didn’t see a substrate here; you may have seen a PRODUCT presumably ready to be released, as the nature of the lipoamide state was not clear. Also, it should be mentioned that the “other” substrate (CoA) is not present in this study, although a model was utilized from a related structure to provide insight. That’s certainly appropriate in this case, but should be made clear early on.

We have changed the text to “likely inserts”, as requested and added the information about the absence of coA as follows:

“We report here the cryo-EM reconstruction of the cubic E2p core of E. coli PDHc in a native resting state (in the absence of substrates), in which a lipoyl domain is bound to the catalytic domain of each E2p and the dihydrolipoyllysine residue is immersed deep in the E2p active site pocket (coenzyme A is not present).”

Lines 163 and 164, I would change “of the E2p core” to “of the entire complex.” It’s too confusing with both the words “octahedron” and “cubic” used in the same sentence to describe the core, and it’s only the core thought to be cubic.

We have changed the text as requested.

Lines 192 & 193, It’s well known that in the absence of substrate (pyruvate) E1 does NOT bind LD, regardless of the lipoamide state. I think there may be serious errors in the description/assessment of various lipoamide states abilities to bind to the components in this and the next few lines. If I’m mistaken, references for these assessments are needed here as they are not obvious or common knowledge, even to those familiar with the field.

Thank you for this comment. There is evidence in the literature, that E1p (and also E1o) from E. coli are capable of binding lipoylated LDs also when pyruvate is not present, although with a lower affinity. We have included the reference (Jones, D. D., Stott, K. M., Reche, P. A. & Perham, R. N. Recognition of the lipoyl domain is the ultimate determinant of substrate channelling in the pyruvate dehydrogenase multienzyme complex. J. Mol. Biol. 305, 49-60 (2001).) and reformulated the sentences as follows:

“The reduced form may bind to both E2p and E3, while the oxidized form may bind to both E1p and E3 (Fig. 1a), and the reduced form likely has a higher affinity towards E2p than the oxidized form. Due to the lack of pyruvate in the cells, the PDHc exists in a resting state, where E1p is not active. Lipoylated LDs have been shown to bind to E1p (and also to E1o) with a lower affinity also in the absence of pyruvate⁴⁷ (resting state). This work⁴⁷ was done using only isolated domains and we believe that in the context of the whole complex, given the high local concentration of covalently tethered LDs, the binding to E1p would be efficient even at the low affinity. We therefore assess that E1p might be occupied by the trapped LDs with oxidized lipoyllysine. Therefore, other LDs carrying oxidized lipoyllysine (the product of E3) would remain bound to E3 and thus block the E3 active site from the access of the substrate, which are the LDs with reduced dihydrolipoyllysine. These LDs with dihydrolipoyllysine could thus remain bound to the active site of E2p, as the product of E2p (Fig. 1a).”